# Weak-eval-Strong: Evaluating and Eliciting Lateral Thinking of LLMs with Situation Puzzles

**Qi Chen  Bowen Zhang  Gang Wang  Qi Wu**[†]
Australian Institute for Machine Learning, University of Adelaide
{qi.chen04, b.zhang, qi.wu01}@adelaide.edu.au, gang@theb.ai

## Abstract

While advancements in NLP have significantly improved the performance of Large Language Models (LLMs) on tasks requiring vertical thinking, their lateral thinking capabilities remain under-explored and challenging to measure due to the complexity of assessing creative thought processes and the scarcity of relevant data. To address these challenges, we introduce SPLAT, a benchmark leveraging Situation Puzzles to evaluate and elicit LAteral Thinking of LLMs. This benchmark, containing 975 graded situation puzzles across three difficulty levels, employs a new multi-turn player-judge framework instead of the traditional model-based evaluation, which often necessitates a stronger evaluation model. This framework simulates an interactive game where the model (player) asks the evaluation model (judge) questions about an incomplete story to infer the full scenario. The judge answers based on a detailed reference scenario or evaluates if the player's predictions align with the reference one. This approach lessens dependence on more robust evaluation models, enabling the assessment of state-of-the-art LLMs. The experiments demonstrate that a robust evaluation model, such as WizardLM-2, closely matches human judgements in both intermediate question-answering and final scenario accuracy, achieving over 80% agreement–similar to the agreement levels among humans. Furthermore, applying data and reasoning processes from our benchmark to other lateral thinking-related benchmarks, *e.g.*, RiddleSense and BrainTeaser, leads to performance enhancements. This suggests that our benchmark effectively evaluates and elicits the lateral thinking abilities of LLMs. Code is available at: https://github.com/chenqi008/LateralThinking.

## 1  Introduction

Vertical and lateral thinking are two essential styles that play critical roles in human cognition and decision-making [42]. As noted in [20], vertical thinking, characterised by its logical and structured nature, involves a systematic, step-by-step approach to problem-solving where each step logically follows the previous one. In contrast, lateral thinking is about creativity and viewing problems from multiple angles. It involves breaking away from traditional thought patterns to generate new ideas, and embracing a more playful and imaginative problem-solving approach.

The evolution of natural language processing (NLP) models, particularly large language models (LLMs) [10, 15, 40, 41], has seen significant advancements in tasks that require vertical thinking, such as complex reasoning [5, 44] and commonsense inference [6, 38]. Despite these achievements, the development and evaluation of these models have primarily focused on vertical thinking capabilities [7, 43], often neglecting lateral thinking, which is essential for creatively solving novel problems. Current benchmarks [35, 36] frequently dismiss creative thinking as irrelevant, focusing only on

---

[†]Corresponding author.

38th Conference on Neural Information Processing Systems (NeurIPS 2024).

Figure 1: Examples from SPLAT benchmark, which are categorised into three ascending levels of difficulty, *i.e.*, Easy, Medium, and Hard. The puzzles that are medium to hard in difficulty usually require guidance from the judge to be solved. If a puzzle can be solved without the judge's guidance, it typically resembles a regular puzzle requiring specific knowledge more than lateral thinking skills.

problems solvable through conventional commonsense reasoning, indicating a gap in traditional LLMs' handling of unconventional thinking tasks, such as puzzles [20].

To fill this gap, benchmarks increasingly aim for tasks challenging even for humans, with outputs that are typically open-ended [8, 27]. However, evaluating such open-ended generation often relies on human or model-based assessments [47]. As tasks grow in complexity, such as longer output requirements, human evaluations become less feasible due to the extensive time and effort required to accurately judge long, complex scenarios created by LLMs. Moreover, model-based evaluations typically depend on more advanced models, thereby restricting their capacity to evaluate newer, state-of-the-art models. This reliance may also introduce biases, such as a preference for the first choice [47]. Consequently, there is a pressing need to rethink the evaluation framework of the benchmark for assessing emerging LLMs.

In this paper, we seek to explore and elicit the lateral thinking ability of LLMs. However, accurately evaluating this capability poses significant challenges due to the complexity of measuring creative thinking [29, 19] and the difficulty of obtaining relevant data. The generation of novel ideas is inherently non-trivial, even for humans [13, 14]. Considering these challenges, we propose the exploration of lateral thinking in LLMs by *situation puzzles* as a primary research tool. A situation puzzle, often referred to as a lateral thinking puzzle, involves a scenario, usually presented as an unusual situation, and the goal is to figure out what is going on. Players ask yes-or-no questions to gather more information and solve the puzzle. These puzzles challenge players to think outside the conventional bounds of the scenario's information, using creativity and indirect reasoning to uncover the underlying explanation for the situation described. In this context, we introduce SPLAT, a benchmark that leverages Situation Puzzles for evaluating and eliciting LAteral Thinking of LLMs, which contains 975 high-quality situation puzzle pairs. As shown in Figure 1, for better assessment, we categorise these puzzles into three difficulty levels—Easy, Medium, and Hard—each annotated by human evaluators. We obtain 54,463 annotations of difficulty on an average of ∼55 per puzzle.

To assess the performance of LLMs in our benchmark with an open-ended setting, we initially aim to apply a model-based evaluation paradigm. However, this paradigm (*e.g.*, [47]) typically relies on using an evaluation model that is stronger than the models being tested. This creates a challenge in accurately evaluating models that may be superior to the evaluator itself. Therefore, there is a need for alternative methods that can effectively assess the advanced models without this limitation. To this end, inspired by the situation puzzle game, we propose a multi-turn player-judge framework. This framework emulates an interactive game where the model (player) queries the evaluation model (judge) about a given incomplete story, striving to collect information to unravel the intricate scenario it implies. Conversely, the evaluation model (judge) is tasked solely with responding to questions based on the detailed reference scenario or evaluating whether the scenario deduced by the player aligns semantically with the reference one[1]. In this way, this framework reduces the reliance on more advanced evaluation models, and thereby can be used to evaluate the state-of-the-art LLMs.

The experimental results indicate that a relatively strong evaluation model, *e.g.*, WizardLM-2, can align closely with human judgements in both intermediate question-answering and final answer/scenario accuracy, achieving over 80% agreement—comparable to agreement level between

---

[1] Note that the ground-truth reference answer/scenario only can be seen by the judge.

humans. Moreover, applying the data and reasoning processes from our benchmark to other lateral thinking-relevant benchmarks like RiddleSense [22] shows a performance improvement (*e.g.*, the accuracy of Llama3-70B [1] increase from 83.34% to 85.21%). It indicates that our benchmark can not only evaluate but also elicit the lateral thinking capabilities of LLMs. Finally, we evaluate several advanced LLMs such as GPT-4 and GPT-4 Turbo on our benchmark. The results highlight the ongoing challenges and the need for improved lateral thinking in LLMs.

We summarise our contributions as follows:

- We introduce a benchmark, called SPLAT, to evaluate the lateral thinking capacity of LLMs with situation puzzles. In this benchmark, we construct a new situation puzzles dataset, containing 975 unique puzzles that are divided into three difficulty levels.
- We design a multi-turn player-judge evaluation framework to move beyond the existing model-based evaluation paradigm in open-ended settings. This new framework reduces reliance on a stronger evaluation model – typically stronger than the models being assessed.
- The dataset and framework are designed not only to evaluate but also to actively elicit the lateral thinking of LLMs. Experiments show that using data and reasoning processes from our framework leads to improved performance of LLMs, even when applied to other lateral thinking benchmarks.

## 2   Related Works

**Lateral Thinking for LLMs.** The realm of lateral thinking and computational creativity spans various tasks [50, 26, 48, 39, 18, 32], such as pun detection [50] and humour recognition [26, 48], which assess different cognitive capabilities. Notable contributions in this field include BrainTeaser [20], which evaluates a broad spectrum of human intelligence attributes such as strategy and creativity. Another significant effort is RiddleSense [22], a collection of riddles from public websites aimed at testing model abilities. In contrast, our work enhances this area by introducing scenario-level lateral thinking puzzles, *i.e.*, situation puzzles, constructing a benchmark to evaluate and elicit the lateral thinking ability of Large Language Models (LLMs).

**Benchmarks for LLMs.** Large language models (LLMs) [2, 3, 7, 37] are increasingly showing capabilities for various tasks ranging from writing and coding to engaging in multi-turn dialogues. However, assessing their extensive capabilities presents new challenges. Existing benchmarks, as categorised by [47], predominantly fall into three types. Core-knowledge benchmarks [17, 46, 11, 34, 9, 12, 49] assess LLMs through zero-shot and few-shot tasks, requiring concise, specific answers to questions that are easily validated. Instruction-following benchmarks [24, 28] handle more open-ended questions and a broader range of tasks, which evaluate LLMs post-instructional fine-tuning. Lastly, conversational benchmarks [30, 16, 21], designed for dialogue-based tasks, still lack the diversity and complexity needed to fully challenge advanced chatbots.

For more challenging benchmarks, there is a trend toward including tasks that are even more difficult for humans, often requiring open-ended outputs [8, 27]. Evaluating such open-ended tasks typically depends on either human or model-based assessments. As these tasks increase in complexity, human evaluations become impractical due to the substantial time and effort needed to accurately assess the lengthy and intricate scenarios. Existing model-based evaluations [47] inherently rely on stronger models, which limits their ability to assess new state-of-the-art models. In this paper, we introduce a multi-turn player-judge framework to improve upon traditional model-based evaluations in open-ended settings. It reduces the need for a stronger evaluation model typically used in such assessments.

## 3   Task and Dataset

### 3.1   Task Definition of Situation Puzzle Game

The situation puzzle game involves a player $P$ and a judge $J$. Firstly, the judge provides an incomplete story $S_0$ to the player. Then, the player poses a set of questions $\mathcal{Q} = \{q_1, q_2, \ldots, q_i, \ldots\}$ gradually for the judge to gather information about the unknown and detailed scenario/answer $\hat{S}$ behind the initial story $S_0$. The judge responds with answers $\mathcal{A} = \{a_1, a_2, \ldots, a_i, \ldots\}$, where each answer $a_i \in \{\text{yes}, \text{no}, \text{irrelevant}\}$. The objective is to deduce the hidden scenario/answer $\hat{S}$ that semantically aligns with all preceding yes-or-no question-answering pairs while fitting the given story $S_0$.

Table 1: Comparison between our SPLAT and related benchmarks, including both vertical (Ver.) and lateral (Lat.) thinking. For a comprehensive comparison, we evaluate the benchmarks across four dimensions: the target task, the fashion of answers, the evaluation paradigm, and the statistics of samples.'Ref. A' indicates whether the question or dialogue comes with a reference answer. 'Open-E.' denotes whether the response is open-ended. 'Ref.-G.' and 'Mod.-B.' refer to whether the result evaluation is reference-guided or model-based, respectively. 'Assessment' denotes the specific evaluation pattern used to assess the quality of the predicted results. 'Avg. Q' and 'Avg. A' are the average number of tokens in questions (or input dialogues) and reference answers, respectively.

| | Benchmark | Task | Answer | | Evaluation | | | Statistic | | |
| | | | Ref. A | Open-E. | Ref.-G. | Mod.-B. | Assessment | Number | Avg. Q | Avg. A |
|---|---|---|---|---|---|---|---|---|---|---|
| Ver. | GAIA [27] | Question-Answering | ✓ | ✓ | ✓ | | Exact Match | 466 | 65.85 | 2.03 |
| | Chatbot Arena [47] | Dialogues | | ✓ | | ✓ | Battle | 30,000 | / | / |
| | MT-Bench (1st Turn) [47] | Multi-turn QA | | ✓ | | ✓ | Grading or Battle | 80 | 65.78 | / |
| | MT-Bench (2nd Turn) [47] | Multi-turn QA | | ✓ | | ✓ | Grading or Battle | 80 | 22.76 | / |
| Lat. | Oogiri (T2T, Eng.) [48] | Multiple-choice QA | ✓ | | ✓ | | Ranking | 6,433 | 12.46 | 10.38 |
| | RiddleSense [22] | Multiple-choice QA | ✓ | | ✓ | | Ranking | 5,733 | 26.21 | 2.87 |
| | BrainTeaser (S.) [20] | Multiple-choice QA | ✓ | | ✓ | | Ranking | 627 | 34.88 | 9.11 |
| | BrainTeaser (W.) [20] | Multiple-choice QA | ✓ | | ✓ | | Ranking | 492 | 10.65 | 3.00 |
| | SPLAT (Ours) | Scenario Deduction | ✓ | ✓ | ✓ | ✓ | Semantic Judge | 975 | 79.74 | 45.53 |

## 3.2 Data Construction

**Data Collection.** To collect the situation puzzle data, we follow the steps from [20]. Specifically, we first gather over one thousand situation puzzles and their answers from various public websites using web crawlers[2]. After that, we merge the data from different sources and then eliminate the duplicates based on sentence similarity [31]. Then, we correct typographical errors using the Auto Correct library[3], followed by a human review process to ensure the puzzles maintain their intended meanings. This approach combines automated corrections with manual oversight to preserve the quality of the puzzles. Finally, we obtain a collection of 975 unique situation puzzles.

**Data Annotation and Difficulty.** Inspired by the organisation of GAIA [27], we categorise our collected situation puzzles into various distinct levels of difficulty, ranging from 1 (easiest) to 9 (hardest). This classification is done through a crowd-sourced annotation involving an average of approximately 55 annotators per puzzle to ensure the reliability of the difficulty rating. Then, these levels are further grouped into three broader categories: Easy (1∼3), Medium (4∼6), and Hard (7∼9).

## 3.3 SPLAT Benchmark (Ours) vs. Related Benchmarks

We compare our benchmark with the most relevant ones *w.r.t.* both vertical and lateral thinking. Specifically, for vertical thinking datasets, we consider GAIA [27] and LLM-as-a-Judge [47], where the former is the most recent benchmark for general AI assistants, and the latter is the widely used benchmark to evaluate LLMs. The LLM-as-a-Judge [47] contains two different benchmark datasets, *i.e.*, MT-bench and Chatbot Arena. The MT-bench is a set of challenging multi-turn open-ended questions designed to evaluate chat assistants. Chatbot Arena is a benchmark platform designed for evaluating LLMs through interactive, randomised battles in a crowd-sourced manner. It allows for the assessment of LLMs in real-time dialogues with human participants.

Regarding lateral thinking benchmarks, we compare Oogiri [48], RiddleSense [22], and Brain-Teaser [20]. As the Oogiri benchmark [48] is multi-modal (text and image) and multi-lingual (English, Chinese, and Japanese), for a fair comparison, we consider Text-to-Text (T2T) English (Eng.) samples only. RiddleSense [22] is a benchmark dataset designed for evaluating complex commonsense reasoning within the framework of riddle-style questions. It presents a unique multiple-choice question-answering task that requires an understanding of figurative language, counterfactual reasoning, and other advanced natural language understanding skills. The BrainTeaser dataset [20] is a versatile and challenging collection for evaluating models on their ability to engage in lateral thinking and solve intricate puzzles. It features two main types of puzzles: Sentence (S.) puzzles and Word (W.) puzzles. The sentence puzzles require solvers to interpret or decipher complex sentences, often involving multiple meanings or requiring the solver to think beyond straightforward interpretation.

---

[2]Upon publication, we will release our data, as all the puzzles are sourced from publicly accessible websites.
[3]https://github.com/phatpiglet/autocorrect

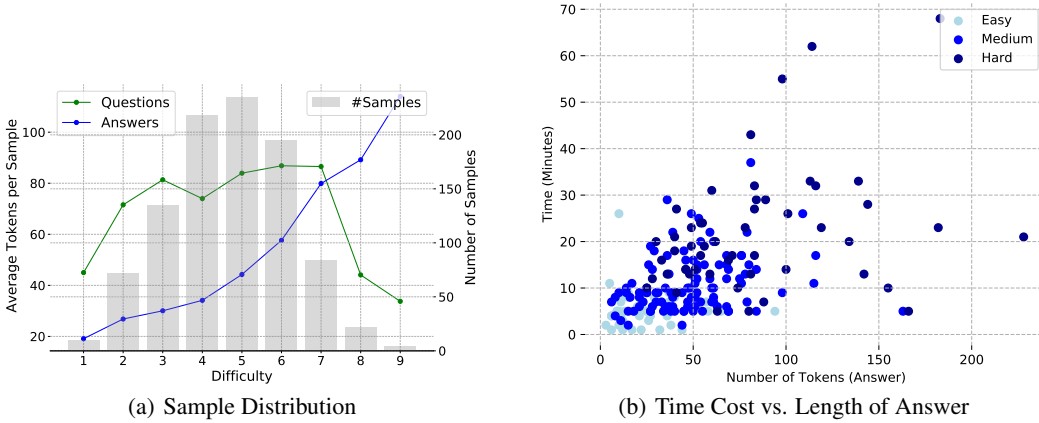

(a) Sample Distribution

(b) Time Cost vs. Length of Answer

Figure 2: We show (a) the distribution of sample counts across different difficulty levels, and the average number of tokens per sample. We also exhibit (b) the distribution of time cost for human players and the number of tokens for each reference answer (200 samples in total) in difficulty levels.

The word puzzles focus on wordplay or semantic puzzles, where the challenge lies in understanding and manipulating words based on how they are presented or their semantic relationships.

In Table 1, we compare our SPLAT benchmark with existing ones across various dimensions such as the type of target task, the fashion of answers, the evaluation paradigm, and the statistics of samples. Here's a detailed discussion of how our benchmark, focused on situation puzzles, stands out:

**Task Uniqueness and Open-ended Responses.** Unlike traditional QA and dialogue-based tasks in [27, 47], our task requires models to infer the scenario from a given incomplete story, necessitating deep engagement with both context and creativity. The process of accurately inferring the scenario involves multiple steps and is challenging to brute force due to their diversity [27], as well as difficult to evaluate. Our benchmark supports the verification of reasoning traces and predicted answers. Moreover, unlike the multiple-choice formats seen in tasks like Oogiri [48] and RiddleSense [22], our open-ended format lessens the likelihood of inadvertently leading to the correct answer.

**Reference-guided Semantic-aware Evaluation.** Unlike the exact match or pairwise battle approaches used in vertical thinking benchmarks [47, 27], our SPLAT employs a 'Semantic Judge' evaluation. Analogous to Natural Language Inference (NLI) [25], we assess the semantic alignment between the predicted output and a human-generated reference answer using an LLM-based judge. This approach ensures accurate evaluation, even when the formats of the predicted outputs significantly differ from those of the reference answers.

**Challenging Question with Judgement-friendly Answer.** Our dataset features 975 samples with relatively high averages in question and answer lengths (79.74 and 45.53 tokens respectively). This indicates a depth of content and complexity in our scenarios, surpassing most lateral thinking datasets such as RiddleSense [22] or BrainTeaser [20], where the questions or answers are typically much shorter and less detailed. Moreover, by employing a model-based and reference-guided evaluation paradigm, we reduce the dependence on a more robust evaluation model, different from [47], which typically needs a stronger model to assess. Our approach only necessitates that the judge model determines if the predicted answer is semantically aligned with the reference answer. Once we verify that the used judge model can reliably perform this task, the relative strength of the models being evaluated compared to the judge model becomes irrelevant.

### 3.4 Statistics on SPLAT Benchmark

Situation puzzles on the proposed SPLAT can be classified into three ascending levels of difficulty, based on three criteria. 1) *Time to Solve:* How long does it typically take for a player to arrive at a correct answer? Shorter times indicate easier puzzles, while longer time suggests higher complexity. 2) *Length of Reference Answer/Scenario:* Simpler puzzles may have shorter, more straightforward answers, whereas more complex ones might require lengthy explanations or involve intricate scenarios.

3) *Subjective Complexity of the Scenario:* This involves a subjective evaluation of how mentally challenging the scenario is for the human players. Thus, the puzzle in each level can be defined as:

**Easy Puzzles** (1∼3): These puzzles typically require minimal time to solve, with reference answers that are short and direct, involving straightforward solutions. These scenarios are simple and can usually be resolved quickly, often within a few minutes, and do not necessitate extensive reasoning.

**Medium Puzzles** (4∼6): These are moderately complex puzzles that demand more time and deeper analysis. The reference answers are longer, providing more details that players must consider. Solving these puzzles generally takes a moderate amount of time, with players needing to make several logical deductions and possibly sift through some distractors.

**Hard Puzzles** (7∼9): These are the most challenging puzzles, designed for advanced players. They require a significant amount of time to unravel, with lengthy and complex scenarios that may include extensive details or embedded subtleties. Players must engage in a high degree of reasoning, with an array of logical deductions and the potential for numerous misleading clues. These puzzles often demand an arbitrary number of steps, deep thinking, and the ability to focus over prolonged periods.

Figure 2(a) suggests a correlation between difficulty level and the average number of tokens, particularly for answers, which tend to increase in length as the difficulty level rises. This could indicate that more complex puzzles require longer, more detailed answers. In Figure 2(b), the clusters by difficulty level also show that harder puzzles tend to have both longer answers and higher time costs, aligning with intuitive expectations about puzzle solving.

## 4   Multi-turn Player-Judge Framework

We design such a framework (Figure 3) for three main reasons. Firstly, the complexity of scenarios in situation puzzles makes it difficult for a model to accurately predict the final answer in just one attempt (*i.e.*, in a single turn). Secondly, we aim to ensure that the reasoning process remains interpretable and assessable, even in the absence of pre-defined reference reasoning steps. This framework allows for a more thorough evaluation of the model's ability to solve complex puzzles step by step. Thirdly, we hope that the framework or the intermediate reasoning processes derived from this framework can help elicit the lateral thinking ability of LLMs even on other lateral thinking-relevant datasets.

**Characters.** The multi-turn player-judge framework of a situation puzzle game has two primary characters: a player $P$ and a judge $J$. The player $P$ is a participant (*e.g.*, LLMs we need to evaluate), who attempts to solve the puzzle by asking questions. The judge $J$ is the entity, either a human or an AI model, that knows the solution to the puzzle and provides responses to the player's questions. Due to the page limit, we put the detailed character guidelines in the supplementary.

**Interaction Dynamics.** The interaction between players and the judge is driven by a series of questions $\mathcal{Q} = \{q_1, q_2, \ldots, q_i, \ldots\}$ and answers $\mathcal{A} = \{a_1, a_2, \ldots, a_i, \ldots\}$. Specifically, at the start of the game, the player is presented with an incomplete story, denoted as $S_0$. The player must read this story and initially decide if the hidden scenario or answer $\hat{S}$ can be inferred from the present information (*i.e.*, solely $S_0$). If the scenario remains unclear, the player will then pose a yes-or-no question $q_1$ to obtain more details about the puzzle.

The judge provides a response $a_1 \in \{\text{yes}, \text{no}, \text{irrelevant}\}$ based on the incomplete story $S_0$ and the hidden scenario $\hat{S}$[4], *i.e.*, $a_1 \leftarrow J(q_1, S_0, \hat{S})$. Upon receiving the response $a_1$, the player will either attempt to deduce the final answer or, if still uncertain, will ask a subsequent question (*i.e.*, $q_2$). This process continues iteratively, with the player refining the understanding based on the judge's answers.

**Information Sets and Game State.** At any turn $t$ in the game, the information set $\mathcal{I}_t$ represents all the questions asked and answers received up to that time:

$$\mathcal{I}_t = \{(q_1, a_1), (q_2, a_2), \ldots, (q_t, a_t)\}. \tag{1}$$

The game state $\Gamma_t$ includes the incomplete story $S_0$, the information set $\mathcal{I}_t$ and any inferred knowledge $K_t$ about the situation puzzle:

$$\Gamma_t = (S_0, \mathcal{I}_t, K_t), \text{ where } K_t \leftarrow P(S_0, \mathcal{I}_t). \tag{2}$$

---

[4]Note that the judge knows the detailed scenario, *i.e.*, the reference answer.

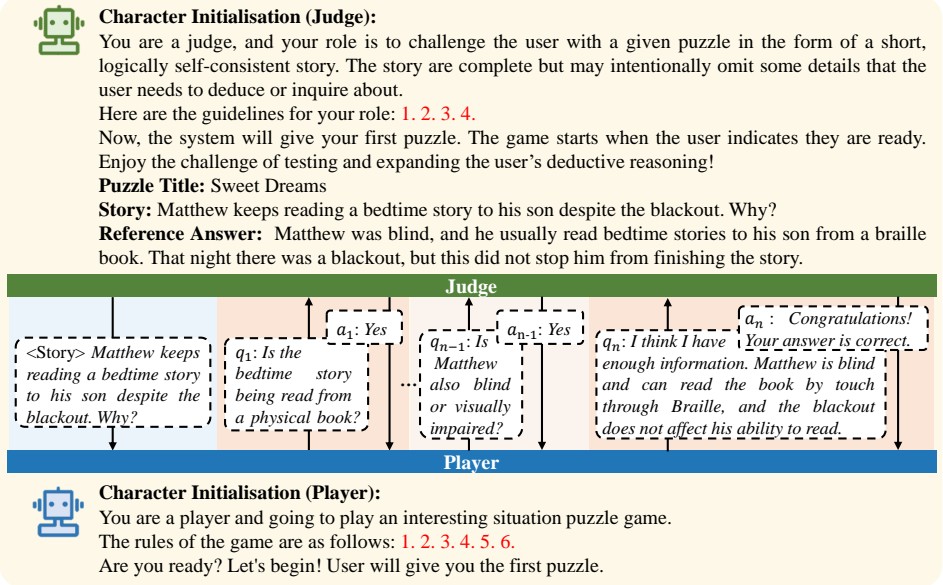

Figure 3: Overall of the multi-turn player-judge framework. The player begins with a given story and poses yes/no questions to uncover hidden details. The judge, informed by a reference answer, responds to guide the player toward the correct reasoning. The player's goal is to deduce the scenario based on the judge's feedback and the initial story input. The game continues with questions until the player deduces the correct answer, at which point the judge confirms with a congratulatory response.

Here, $K_t$ is the knowledge inferred by the player based on the story $S_0$ and information set $\mathcal{I}_t$. The player determines the next question based on the current game state by

$$q_{t+1} = P(\Gamma_t). \tag{3}$$

**Objective.** The objective of the player is to deduce the scenario/answer $S$ that satisfies all given constraints based on the responses from the judge. Mathematically, the goal is to find the hypothesised solution $\hat{S}$ that maximises the posterior probability given the game state $\Gamma_t$:

$$\hat{S} \leftarrow \arg\max_S \Pr(S|\Gamma_t). \tag{4}$$

## 5 Experiments

### 5.1 Agreement Evaluation

Inspired by [47], to assess the alignment of judge's responses with human preferences, we consider an agreement evaluation among them *w.r.t.* judgements for both final predicted answers and the intermediate reasoning processes. We randomly select 20 puzzles from each difficulty level, which is 60 puzzles in total. For each puzzle, we ask three different individuals to assess the final answer agreement and the reasoning process agreement, respectively. For a fair comparison, we use the questions generated by taking Llama3 (70B-Instruct) [1] as the player and WizardLM-2 (8x22B) [45] as the judge. Besides humans, we also consider Llama3 (70B-Instruct) as another judge model to test the agreement among humans and different judge models. More details are in the supplementary.

**Final Answer Agreement.** To quantitatively evaluate the semantic alignment between the final predicted answers and reference answers, we first ask humans to assess the semantic alignment between each predicted answer and the corresponding reference answer/scenario. Subsequently, we measure the agreement performance between the human judgement and the evaluations from the judge model. We totally obtain 156 judgements from humans. In Table 2, the high level of agreement between the LLM-based judges and humans (*e.g.*, WizardLM-2 achieves 100%, 87.50%, 88.24% agreements with humans in three difficulties, respectively) indicates that LLMs can well align with human standards of reasoning and judgement within this specific context of lateral thinking puzzles.

Table 2: Final answer agreement among three types of judges on SPLAT benchmark.

| Judge Model | Easy | | Medium | | Hard | |
|---|---|---|---|---|---|---|
| | Llama3-70B | Human | Llama3-70B | Human | Llama3-70B | Human |
| WizardLM-2 | 93.21% | 100% | 92.49% | 87.50% | 90.02% | 88.24% |
| LLama3-70B | - | 84.56% | - | 80.61% | - | 83.17% |
| Human | - | 100% | - | 87.50% | - | 100% |

Table 3: Reasoning process agreement among three types of judges on SPLAT benchmark.

| Judge Model | Easy | | Medium | | Hard | |
|---|---|---|---|---|---|---|
| | Llama3-70B | Human | Llama3-70B | Human | Llama3-70B | Human |
| WizardLM-2 | 82.15% | 84.18 % | 78.82% | 89.86% | 72.14% | 90.15% |
| LLama3-70B | - | 80.22% | - | 77.83% | - | 71.36% |
| Human | - | 92.93% | - | 94.13% | - | 93.11% |

**Reasoning Process Agreement.** To assess the agreement on reasoning processes, particularly for intermediate questions where no predefined reference answers exist, we use a method where humans are asked to respond to these questions based on the context provided by the given story and the final reference answer. Their responses are categorised as {yes, no, irrelevant}. We then evaluate the alignment of these human responses with those from the judge model. We obtain 2,271 judgements from humans. Table 3 shows that LLMs, *e.g.*, WizardLM-2 and Llama3-70B, demonstrate good alignment with human reasoning in simpler puzzles. As the difficulty level rises, WizardLM-2 keeps a relatively high agreement with human judgements, while the alignment between Llama3-70B and humans decreases dramatically. The results verify that our choice of WizardLM-2 as the judge model aligns well with human judgement across all difficulty levels, maintaining over 80% agreement.

## 5.2 Performance of LLMs on SPLAT Benchmark

**Evaluation Metrics.** To assess the performance of LLMs using the proposed SPLAT benchmark, we employ two distinct metrics: *Accuracy (Acc, %)*, which measures the correctness of the scenarios deduced by LLMs against the reference scenarios/answers, and *Average Round (Rnd)*, which quantifies the average number of interaction rounds required for LLMs to reach the reference scenarios. Note that if the LLMs fail to deduce the correct scenario before a pre-defined max round, the round count is the same as the max one, and the accuracy for that particular puzzle is recorded as 0. Conversely, if the correct scenario is successfully predicted, the accuracy is set to 1, and the number of rounds taken to reach the correct answer is recorded. Finally, we define an *OverAll (O/A)* evaluation metric as

$$\text{O/A} = \frac{1}{N} \sum_{i=1}^{N} \frac{\mathbb{I}(\text{sample}_i)}{\text{Rnd}_i} \times 100, \tag{5}$$

where $\mathbb{I}(\cdot)$ is an indicator that returns 1 if the scenario deduced by LLMs matches the reference one semantically, and 0 otherwise. Given that $\mathbb{I}(\text{sample}_i)$ takes values in {0, 1} and $\text{Rnd}_i$ ranges from 1 to a maximum, the O/A metric spans from 0 to 100, with higher values indicating better performance.

Besides, in the open-ended setting, the player model may deduce other reasonable scenarios that satisfy both the given incomplete story and the responses from the judge model. In this situation, it is hard to judge whether these deduced scenarios are correct or not due to the diversity. Inspired by the evaluation of image captioning on MSCOCO [23], which uses multiple reference answers for each image and computes metrics based on the closest reference answer, we consider a similar approach. However, generating multiple reference scenarios/answers for situation puzzles is challenging. As an alternative, we repeatedly run each situation puzzle multiple times (denoted as $R$), and then choose one (maybe $r$-th) that is closest to the reference scenario from these iterations as the final result. This approach not only maintains the diversity of potential predictions but also reduces the impact of hallucination from LLMs by subjecting each puzzle to several evaluations.

**Results.** We compare the performance of various LLMs on our SPLAT benchmark, including Lama3 (8B-Instruct & 70B-Instruct) [1], Qwen1.5 (32B-Chat & 110B-Chat) [4], WizardLM-2 (8x22B) [45], GPT-4 and GPT-4 Turbo [2]. Here, we take these LLMs as players and use WizardLM-2 (8x22B) as a judge in all these performance comparison experiments. We set $R = 1$ and the max round is 15. In Table 4, GPT-4 and its Turbo variant, along with the Llama3 (70B-Instruct) and WizardLM-2 (8x22B), show robust capabilities, achieving relatively higher accuracy. Conversely, models like

Table 4: Performance of different LLMs on the proposed SPLAT benchmark.

| | Easy | | | Medium | | | Hard | | | Average | | |
|---|---|---|---|---|---|---|---|---|---|---|---|---|
| | Acc (↑, %) | Rnd (↓) | O/A (↑) | Acc (↑, %) | Rnd (↓) | O/A (↑) | Acc (↑, %) | Rnd (↓) | O/A (↑) | Acc (↑, %) | Rnd (↓) | O/A (↑) |
| Llama3-8B | 31.79 | 13.32 | 3.65 | 14.81 | 14.29 | 1.58 | 4.55 | 14.80 | 0.50 | 17.05 | 14.13 | 1.91 |
| Llama3-70B | 64.05 | 10.75 | 8.67 | 28.54 | 13.37 | 3.54 | 10.91 | 14.31 | 1.36 | 34.50 | 12.81 | 4.52 |
| Qwen1.5-32B | 29.95 | 12.72 | 6.39 | 15.58 | 14.11 | 2.47 | 10.00 | 14.22 | 2.01 | 18.51 | 13.68 | 3.62 |
| Qwen1.5-110B | 46.08 | 11.28 | 9.71 | 24.22 | 13.48 | 3.74 | **17.27** | 14.01 | 2.16 | 29.19 | 12.92 | 5.20 |
| WizardLM-2 | 58.52 | 10.49 | 11.97 | 31.48 | 13.12 | 4.67 | 16.36 | **13.96** | **2.74** | 35.45 | 12.52 | 6.46 |
| GPT-4 Turbo | 60.36 | 9.59 | 13.22 | 28.39 | 12.87 | 5.06 | 10.00 | 14.37 | 1.35 | 32.91 | 12.27 | 6.54 |
| GPT-4 | **70.96** | **8.93** | **15.25** | **36.88** | **12.41** | **6.71** | **17.27** | 13.99 | 2.52 | **41.70** | **11.77** | **8.16** |

Qwcn1.5 (32B-Chat) and Qwcn1.5 (110B-Chat) show lower accuracy, which might reflect challenges in adapting their chat-based configurations to the lateral thinking required on the SPLAT benchmark.

**Performance Bias.** In the SPLAT benchmark, concerns about bias when a model like WizardLM-2 (8x22B) evaluates its own performance are mitigated through several key design elements. First, the SPLAT benchmark is designed with a specific framework that focuses on how well each LLM, serving as a player, can reason through situation puzzles to arrive at correct scenarios/answers. The judge's role is tasked solely with responding to questions based on the detailed reference scenario or evaluating whether the scenario deduced by the player aligns semantically with the reference one, which is relatively simple and objective. Besides, the evaluation metrics used (Accuracy and Rounds needed) are also objective, reducing the potential for subjective bias or preference.

From Table 4, WizardLM-2 (8x22B) performs well but not exceptionally in every category. It has competitive but not maximal accuracy rates, and its average round count is not the lowest. It suggests that the WizardLM-2 (8x22B) we use has strong enough capabilities to serve as a judge model. Besides, under our benchmark, the used judge model does not have an obvious preference for itself.

## 5.3 Eliciting Lateral Thinking of LLMs

**Models and Benchmarks.** As previously discussed, our benchmark serves to evaluate the lateral thinking abilities of LLMs and to elicit these capabilities actively. To verify this, we seek to enhance LLMs with the data and reasoning processes from our benchmark, and then evaluate them on other lateral thinking benchmarks (*i.e.*, RiddleSense [22] and BrainTeaser [20]) to observe performance enhancements across various LLMs. Our analysis includes both open-source and closed-source LLMs, specifically Llama3 (Instruct version) and GPT-4, with additional considerations for different model sizes, such as Llama3-8B and Llama3-70B.

**Implementations.** In the zero-shot setting, LLMs are challenged with multiple-choice riddles from these benchmarks, where they must choose answers based solely on the riddles presented. The accuracy of their responses is calculated by comparing them to the correct answers. To effectively integrate our benchmark's data and reasoning

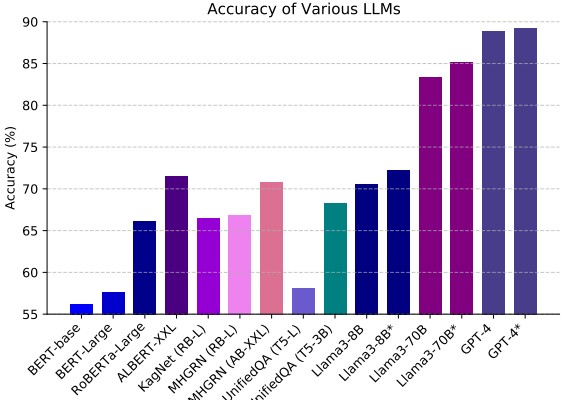

Figure 4: Performance of various LLMs on Riddle-Sense (dev set). Llama3 (8B & 70B) and GPT-4 are in the zero-shot setting, while others are trained on the training set of RiddleSense and CSQA [33]. '*' means models with our auxiliary reasoning prompts.

processes from the proposed player-judge framework, we treat the question-answer pairs generated during the benchmark as auxiliary prompts. These are seamlessly incorporated into the LLMs' reasoning processes to enhance their thinking ability.

To implement this, we set up the game where one of the LLMs (*e.g.*, Llama3-70B) acts as the player with WizardLM-2 continuing as the judge. We randomly select $m$ situation puzzles (we set $m = 1$ in our experiment) from our benchmark dataset and engage both the player and the judge in these selected puzzles using our multi-turn player-judge framework. From each puzzle, we gather the first 5

Table 5: Performance of LLMs on BrainTeaser. Models with '*' mean with our auxiliary reasoning prompts. 'Ori', 'Sem', and 'Con' are 'Original', 'Semantic', and 'Context', respectively. 'OS' means 'Ori&Sem' while 'OSC' is 'Ori&Sem&Con'. 'Overall' is the average of 'Ori', 'Sem', and 'Con'.

| | BrainTeaser (Sentence) | | | | | | BrainTeaser (Word) | | | | | |
| | Instruction-based | | | Group-based | | | Instruction-based | | | Group-based | | |
| | Ori | Sem | Con | OS | OSC | Overall | Ori | Sem | Con | OS | OSC | Overall |
|---|---|---|---|---|---|---|---|---|---|---|---|---|
| LLama3-8B | 70.23 | 63.09 | 69.64 | 52.09 | 38.32 | 67.65 | 47.72 | 44.69 | 46.21 | 31.06 | 17.42 | 46.20 |
| LLama3-8B* | 72.18 | 65.47 | 72.02 | 58.92 | 45.23 | 69.89 | 54.54 | 54.54 | 65.15 | 39.39 | 29.54 | 58.07 |
| LLama3-70B | 89.34 | 87.57 | 86.39 | 84.61 | 75.73 | 87.76 | 71.96 | 71.96 | 69.69 | 62.87 | 49.24 | 71.20 |
| LLama3-70B* | 93.49 | 90.53 | 90.53 | 88.75 | 82.84 | 91.51 | 81.06 | 81.81 | 75.75 | 74.24 | 59.09 | 79.54 |
| GPT-4 | 93.49 | 89.94 | 83.43 | 88.75 | 75.14 | 88.95 | 71.21 | 65.91 | 56.06 | 59.09 | 41.66 | 64.39 |
| GPT-4* | 95.26 | 91.71 | 88.69 | 91.71 | 82.24 | 91.88 | 74.24 | 72.72 | 62.12 | 64.39 | 45.45 | 69.69 |

question-answer pairs to form an auxiliary reasoning prompt set $\mathcal{U}$, comprising a total of $5 \times m$ pairs. This auxiliary prompt set $\mathcal{U}$ is then integrated into the original prompts $\mathcal{P}$ used by various LLMs. It is important to note that the same auxiliary prompts are used across different LLMs during their evaluations on these two benchmarks. This keeps consistency in the supplementary data provided to each model, ensuring a fair comparison of their performance.

**Results on RiddleSense and BrainTeaser Benchmarks.** On the RiddleSense (dev set), Figure 4 shows that with our reasoning prompts, the accuracy of various LLMs consistently improves (Llama3-8B: 70.51% → 72.18%, Llama3-70B: 83.34% → 85.21%, GPT-4: 88.83% → 89.23%) on the RiddleSense benchmark. The results demonstrate that the data and framework in our benchmark can be used to elicit the lateral thinking ability of LLMs when handling other lateral thinking tasks.

BrainTeaser features two main types of puzzles: sentence puzzles and word puzzles. Following its official settings, we assess model performance using two accuracy metrics: 1) *Instance-based Accuracy*: This metric individually evaluates each question—whether it is the original or a reconstructed version. We present instance-based accuracy for both the original puzzles and their semantic and contextual reconstructions. 2) *Group-based Accuracy*: Under this metric, each original puzzle and its variants are treated as a group. The model earns a score of 1 only if it correctly solves all three puzzles within a group. If it fails to do so, the score awarded is 0. From Table 5, in the zero-shot setting, the accuracy of various LLMs consistently improves on BrainTeaser. These demonstrate that the data and framework of our benchmark effectively elicit lateral thinking capabilities of LLMs when applied to various lateral thinking tasks.

Table 6: Impact of our data and reasoning processes. Models with '†' mean using our data only while with '*' mean using both data and reasoning processes. 'RS' refers to the results on RiddleSense (Dev). 'BT (S.)' and 'BT (W.)' are the overall results on BrainTeaser (Sentence) and Brain-Teaser (Word), respectively.

| | RS | BT (S.) | BT (W.) | Avg. |
|---|---|---|---|---|
| Llama3-8B | 70.51 | 67.65 | 46.20 | 61.45 |
| Llama3-8B† | 70.32 | 69.62 | 52.28 | 64.07 |
| Llama3-8B* | 72.18 | 69.89 | 58.07 | 66.71 |
| Llama3-70B | 83.34 | 87.76 | 71.20 | 80.76 |
| Llama3-70B† | 82.95 | 91.12 | 77.27 | 83.78 |
| Llama3-70B* | 85.21 | 91.51 | 79.54 | 85.42 |

**Ablation Study.** We further explore the influence of our data and reasoning processes. In Table 6, incorporating our data into base LLMs leads to an increase in average accuracy (Llama3-8B: 61.45 → 64.07; Llama3-70B: 80.76 → 83.78). This improvement is further enhanced to 66.71 and 85.42, respectively, when we integrate the reasoning processes. These results further demonstrate that both our data and reasoning processes can effectively enhance the lateral thinking capabilities of LLMs.

# 6 Conclusion

In this paper, we address the gap in the lateral thinking capabilities of LLMs despite significant advances in vertical thinking skills. We propose SPLAT, a benchmark using Situation Puzzles to evaluate and elicit LLMs' LAteral Thinking by a multi-turn player-judge framework, which departs from traditional model-based evaluations. This new framework evaluates the LLMs by interacting with a judge model to solve puzzles, reducing the reliance on more robust evaluation models. Our experiments with SPLAT, including a robust evaluation model like WizardLM-2, demonstrate over 80% agreement with human judgements and show improvements when applied to another lateral thinking benchmark-RiddleSense. This highlights SPLAT's effectiveness in evaluating and eliciting lateral thinking in LLMs, suggesting its potential broader application in AI research and development.

**Acknowledgements** This work was supported by the Centre for Augmented Reasoning, an initiative by the Department of Education, Australian Government.

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

# A Appendix / supplemental material

## A.1 Limitations and Social Impacts

**Limitations**  The nature of lateral thinking tasks often encourages non-linear and novel thought processes, which could lead to unpredictable or unexpected outputs from LLMs. While creativity is desired, these outputs might sometimes include harmful, misleading, or dangerously incorrect information, especially if not properly supervised. To alleviate this, we will engage in robust testing across diverse scenarios to understand the boundaries and limitations of the model's creative outputs. Testing should aim to uncover any potential for generating harmful or inappropriate content.

**Broader Impacts**  Our work's advancement in lateral thinking capabilities of Large Language Models (LLMs) could significantly benefit society by enhancing problem-solving abilities across various domains, such as education, engineering, and the arts. By enabling AI to approach problems with a level of creativity akin to humans, these models could provide innovative solutions that defy conventional thinking patterns.

However, the enhancement of LLMs in lateral thinking also brings potential risks, such as the misuse of technology for malicious purposes like fraud or misinformation, job displacement in creative and problem-solving professions, and increased societal dependence on technology for innovative thinking. To counter these risks, it is essential to establish robust ethical guidelines and transparent AI usage policies. We should seek to promote a culture of ethical AI use combined with ongoing education about AI's societal impacts. This will help ensure that the benefits of advanced AI are realised while minimising its potential harms.

## A.2 Character Initialisation

**Judge:**  The guidelines for a judge include:

- **Scenario Understanding**: Read the given short story and the corresponding answer. Make sure you can understand the both story and answer, and their logical relationships.

- **Response**: Respond to the user's questions with only "yes", "no", or "irrelevant". Your answers should help guide the user towards understanding the full context of the puzzle.

- **Evaluation**: Evaluate the user's final answer carefully. If the user's answer is correct, confirm their success and say "Congratulations". If incorrect, allow them to continue asking questions or provide subtle hints to steer them in the right direction.

- **Question Handling**: If there are multiple questions, respond to the first question and ignore others.

**Player:**  The guidelines for a player include:

- Judge will give you a puzzle in the form of a short story that is logically self-consistent. The story will contain all the information needed to solve the puzzle, but some details may be omitted.

- Your task is to ask questions to gather more information and help you find the solution to the puzzle. You can ask any yes/no questions, and the judge will answer truthfully with yes/no/irrelevant.

- During the questioning process, please carefully analyse the known information and clarify the logical relationships between the clues. Don't overlook any seemingly trivial details.

- When you believe you have collected enough information, please provide your answer and explain how you reasoned out the conclusion based on the known clues.

- If your answer is correct, the game ends; if it's incorrect, you can continue asking questions until you find the correct answer.

- You will never ask questions with answers other than yes/no/irrelevant. Remember, the key to the puzzle is to collect clues through reasonable questions and use logical reasoning to draw conclusions.

## A.3   Agreement Evaluation

Inspired by [47], to assess the alignment of judge's responses with human preferences, we consider an agreement evaluation among them *w.r.t.* judgements for both final predicted answers and the intermediate reasoning processes. We randomly select 20 puzzles from each difficulty level, which is 60 puzzles in total. For each puzzle, we ask 3 different individuals to assess the final answer agreement and the reasoning process agreement. We finally obtain 156 judgements from humans for the final answer and 2,271 judgements for the reasoning process.

**Final Answer Agreement**   To quantitatively evaluate the semantic alignment between the final predicted answers and reference answers, we first ask humans to assess the semantic similarity between each predicted answer and the corresponding reference answer. Subsequently, we measure the agreement performance between the human judgement and the evaluations from the judge model. For example, if three humans vote 'matched', 'matched', and 'unmatched' for a puzzle, respectively, the agreement among them, noted as 'human-human', is only $\frac{1}{3}$, as there are three pairs (matched, matched), (matched, unmatched), and (matched, unmatched). If the judge model vote 'matched', the agreement between humans and the judge model is $\frac{2}{3}$.

This process can be mathematically structured as follows:

Define an alignment metric $\mathcal{M}$ that quantifies the agreement between the judge's judgement and human assessments. The metric can be expressed as

$$\mathcal{M} = \frac{1}{m} \sum_{i=1}^{m} \delta(j_i, h_i) \tag{6}$$

where $\delta$ is an indication function that equals 1 if the judgement $j_i \in \{matched, unmatched\}$ from the judge model matches the human judgement $h_i$, and 0 otherwise. Here, $m$ is the total number of comparisons made between the judgements from the judge model and humans, respectively. A higher value of $\mathcal{M}$ indicates a stronger alignment of the decisions of the judge model with human preferences, suggesting that the judgements rendered by the automated system closely mirror those of human evaluators in terms of understanding and evaluating the semantic content of the answers.

**Reasoning Process Agreement**   To assess the agreement on the reasoning process, particularly for intermediate questions where no predefined reference answers exist, we employ a methodology where humans are asked to respond to these questions based on the context provided by the given story and the final reference answer. Their responses are categorised as {yes, no, irrelevant}. We then evaluate the alignment of these human responses with those from the judge model. This process can be mathematically described as follows:

Define a reasoning agreement metric $\mathcal{R}$ to quantify the alignment between the human responses and the judgements provided by the judge model for intermediate questions:

$$\mathcal{R} = \frac{1}{r} \sum_{i=1}^{r} \delta(a_i, u_i) \tag{7}$$

where $\delta$ is an indication function that equals 1 if the human response $u_i$ to the intermediate question matches the judge model's response $a_i$, and 0 otherwise. Here, $r$ represents the total number of intermediate questions evaluated. A higher value of $\mathcal{R}$ indicates a strong alignment between the human responses and the judge model's decisions regarding the intermediate reasoning steps.

**Instruction for Human**   Figures 5 and 6 provide the instruction shown for humans and ask them to write the judgement for the final answer and for the reasoning process, respectively.

## A.4   Examples

We present several examples illustrating the process of situation puzzle games. In these examples, Llama3 (70B-Instruct) [1] serves as the player andWizardLM-2 (8x22B) [45] functions as the judge.

========================================================================

**Puzzle 1:** The title of the puzzle is *Sweet Dreams*. Matthew keeps reading a bedtime story to his son despite the blackout. Why?

Figure 5: Instructions to ask humans to write the judgement for the final answer.

Figure 6: Instructions to ask humans to write the judgement for the reasoning process.

**Interaction Rounds**

**Background:** Read and fully understand the provided short story but do not show the answer to the user.

**Round 1: Player:** Let's start. Here's my first question:
Is the bedtime story being read from a physical book?
**Judge:** Yes.

**Round 2: Player:** That helps! Here's my next question:
Is the blackout caused by a power outage?
**Judge:** Yes.

**Round 3: Player:** Got it! Next question:
Is Matthew reading the story by candlelight or some other alternative light source?
**Judge:** No.

**Round 4: Player:** That helps to rule out some possibilities. Here's my next question:
Had Matthew already started reading the story before the blackout occurred?
**Judge:** Yes.

**Round 5: Player:** That's a useful clue. Here's my next question:
Does Matthew already know the story by heart?
**Judge:** No.

**Round 6: Player:** That helps to rule out another possibility. Here's my next question:
Is the room where Matthew is reading the story already familiar to him, i.e., he has been in the room before the blackout?
**Judge:** Yes.

**Round 7: Player:** That's helpful. Here's my next question:
Is Matthew's son already asleep or at least lying in bed when Matthew continues reading the story despite the blackout?
**Judge:** No.

**Round 8: Player:** Here's my next question:
Is Matthew's son visually impaired or blind?
**Judge:** No.

**Round 9: Player:** That helps to rule out another possibility. Here's my next question:
Is the story being read from a book with raised print or Braille, allowing Matthew to read it by touch?
**Judge:** Yes.

**Round 10: Player:** I think I'm getting close to solving the puzzle. Here's my next question:
Is Matthew also blind or visually impaired?
**Judge:** Yes.

**Round 11: Player:** I think I have enough information to solve the puzzle. Here's my answer:
Matthew keeps reading the bedtime story to his son despite the blackout because he is blind and can read the book by touch through Braille, and the blackout does not affect his ability to read.
**Judge:** Congratulations.

==================================================================

**Puzzle 2:** The title of the puzzle is *Fatal Shot*. A hunter aimed his gun carefully and fired. Seconds later, he realized his mistake. Minutes later, he was dead. Why?

**Interaction Rounds**

**Background:** Read and fully understand the provided short story but do not show the answer to the user.

**Round 1: Player:** Let's start by asking some questions to gather more information. Here's my first question:
Was the hunter aiming at an animal when he fired his gun?
**Judge:** Yes.

**Round 2:  Player:** That helps to narrow it down a bit. Here's my next question:
Was the animal he was aiming at a large game animal, such as a bear or deer?
**Judge:** Irrelevant.

**Round 3:  Player:** That helps to rule out some possibilities. Here's my next question:
Did the hunter die as a direct result of the bullet he fired?
**Judge:** No.

**Round 4:  Player:** That's interesting. So the hunter's death wasn't directly caused by the bullet he fired. Here's my next question:
Was the hunter in a location that was somehow hazardous or dangerous when he fired the gun?
**Judge:** Yes.

**Round 5:  Player:** That's a good clue! With that in mind, here's my next question:
Was the hunter in a location where the sound of the gunshot could cause a problem, such as near a cliff, on a boat, or in a confined space?
**Judge:** Yes.

**Round 6:  Player:** I think I'm starting to get an idea. Here's my next question:
Was the hunter on a cliff or a mountain when he fired the gun, and did the sound of the gunshot cause a rockfall or avalanche that killed him?
**Judge:** Yes.

**Round 7:  Player:** I was hoping that might be the case. Well, I think I have enough information to solve the puzzle now.
Here's my answer:
The hunter aimed his gun carefully and fired, but seconds later, he realized his mistake. The sound of the gunshot caused a rockfall or avalanche, which killed him minutes later.
I reasoned this out by combining the clues that the hunter was aiming at an animal, that he died as an indirect result of firing the gun, and that he was in a hazardous location where the sound of the gunshot could cause a problem. Putting all those clues together led me to the conclusion that the sound of the gunshot triggered a rockfall or avalanche that killed the hunter.
**Judge:** Your answer is on the right track, but it is not fully correct. The key detail you are missing is the specific cause of death. Please continue to ask questions or provide another answer focusing on the exact cause of the hunter's death. Would you like to ask any specific questions to help refine your answer?

**Round 8:  Player:** Thank you for the feedback. I'm glad I was on the right track, but I'd like to refine my answer to get the exact cause of death.
Here's a question to help me refine my answer:
Did the hunter die from being crushed or hit by the falling rocks or debris, or was the cause of death something else related to the rockfall or avalanche?
**Judge:** No.

**Round 9:  Player:** So the hunter didn't die from being crushed or hit by the falling rocks or debris. Here's another question to help me refine my answer:
Did the hunter die from a fall, such as falling off a cliff or down a slope, as a result of the rockfall or avalanche?
**Judge:** No.

**Round 10:  Player:** So the hunter didn't die from a fall either. Here's another question to help me refine my answer:
Was the hunter's death related to a lack of oxygen, such as being buried under snow or trapped in a small space, as a result of the rockfall or avalanche?
**Judge:** Yes.

**Round 11:  Player:** I think I have it now!
Here's my revised answer:
The hunter aimed his gun carefully and fired, but seconds later, he realized his mistake. The sound of the gunshot triggered an avalanche, which buried him under the snow. Minutes later, he died from asphyxiation due to lack of oxygen.
I'm confident that this is the correct answer. The hunter's death was an indirect result of firing the gun, and the specific cause of death was asphyxiation due to being buried under the snow.
**Judge:** Congratulations.

