# OpenReview forum: "Weak-eval-Strong: Evaluating and Eliciting Lateral Thinking of LLMs with Situation Puzzles"
_NeurIPS.cc/2024/Conference — NeurIPS 2024 poster_

### Official Review · Reviewer_eDsP · 2024-06-30

**Soundness:** 3
**Presentation:** 3
**Contribution:** 3
**Rating:** 6
**Confidence:** 4

**Summary:**

The paper constructs the benchmark SPLAT to evaluate and elicit lateral thinking abilities in LLMs. It includes questions similar to brain teasers, which describe scenarios and ask non-obvious questions for guessing.

Example:

__Question/Story__: A hunter aimed his gun carefully and fired. Seconds later, he realized his mistake. Minutes later, he was dead.
__Reference Answer__: He hunted in snow-capped mountains. The shot provoked an avalanche, which covered the man. He died of strangulation.

They also propose a multi-turn player-judge evaluation framework that reduces reliance on stronger evaluation models.

**Strengths:**

1. The SPLAT dataset is different from previous datasets, with graded difficulty levels and harder questions.
2. the multi-turn player-judge framework are claimed to be able to  reduces reliance on stronger evaluation models. (I acutally feel a bit confused for this part, see qustions)
3. The authors conduct experiments to validate their approach, demonstration that SPLAT can improve LLM performance on other lateral thinking tasks suggests broader applicability.

**Weaknesses:**

1. "The dataset and framework are designed not only to evaluate but also to actively elicit lateral thinking in LLMs. Experiments show that using data and reasoning processes from our framework leads to improved performance of LLMs, even when applied to other lateral thinking benchmarks." This part really needs ablation studies. It's unclear which parts are effective. For example, are the reasoning processes, the entire dataset, or similar questions in previous data responsible for the improvement? Did this data improve the model's ability to think about these types of questions, or is the improvement possibly due to some tricks? Moreover, why was testing only done on RiddleSense when many other benchmarks were reported earlier? Is this cherry-picking?

2. A player-judge evaluation framework based on WizardLM-2 is proposed as an alternative to human judges to evaluate the answers, but it shows much worse agreement on hard questions compared to inter-human agreements. However, harder questions are actually proposed as a contribution of this work. As the answers for such puzzles can be "off by a hair but miss by a mile," it might require human evaluation for the hard puzzles. It is also a bit confusing why the human agreement on Medium questions is much lower than on hard and easy ones.

**Questions:**

1. Similar datasets have existed before, as mentioned in the paper, such as RiddleSense and Oogiri. Here's an example from RiddleSense:

My life can be measured in hours. I serve by being devoured. Thin, I am quick; Fat, I am slow. Wind is my foe. What am I? (A) paper (B) candle (C) lamp (D) clock (E) worm

In comparison, SPLAT has longer questions and answers. Also, the reference answers for SPLAT are open-ended, while previous ones are all multiple-choice, which is claimed as a contribution. However, this also makes the evaluation more challenging. What makes you think open-ended is a much better option that makes it a contribution?

2. Author claim that their methods reduces reliance on stronger evaluation models, but it seems they are still using WizardLM-2 as a component (Judge Model) of their framework?

**Limitations:**

The size for this benckmark is small compared with previous similar datasets that have existed before, which include many more puzzles compared to the proposed SPLAT.

---

> ### Author Rebuttal · Authors · 2024-08-07
>
> **W1-1. "The dataset and framework are designed not only to evaluate but also to actively elicit lateral thinking in LLMs. Experiments show that using data and reasoning processes from our framework leads to improved performance of LLMs, even when applied to other lateral thinking benchmarks." This part really needs ablation studies. It's unclear which parts are effective. For example, are the reasoning processes, the entire dataset, or similar questions in previous data responsible for the improvement? Did this data improve the model's ability to think about these types of questions, or is the improvement possibly due to some tricks?**
>
> We conduct an ablation study to examine the impact of our data and reasoning processes on model performance. Besides RiddleSense, we incorporate another lateral thinking benchmark (i.e., BrainTeaser), which includes word and sentence puzzles. The results, as shown in the table below, indicate that incorporating our dataset improves average accuracy (Llama3-8B: from 61.45 to 64.07; Llama3-70B: from 80.76 to 83.78). When we integrate the reasoning processes, these scores further increase to 66.71 and 85.42, respectively.
> These results further demonstrate that our data and reasoning processes can effectively boost LLMs' ability in handling lateral thinking tasks rather than tricks.
>
> | Model | RiddleSense (Dev) | BrainTeaser (Sentence, overall) | BrainTeaser (Word, overall) | Average  |
> | ------- | ------- | ------- | ------- | ------- |
> | Llama3-8B (base) | 70.51 | 67.65    | 46.2 | 61.45 |
> | Llama3-8B (data)| 70.32 | 69.62    | 52.28| 64.07 |
> | Llama3-8B (data+reasoning)  | 72.18 | 69.89    | 58.07| 66.71 |
> | Llama3-70B (base)| 83.34 | 87.76    | 71.2 | 80.76 |
> | Llama3-70B (data) | 82.95 | 91.12    | 77.27| 83.78 |
> | Llama3-70B (data+reasoning) | 85.21 | 91.51    | 79.54| 85.42 |
>
> **W1-2. Why was testing only done on RiddleSense when many other benchmarks were reported earlier? Is this cherry-picking?**
>
> Please refer to General Response **G1**.
>
>
> **W2. A player-judge evaluation framework based on WizardLM-2 is proposed as an alternative to human judges to evaluate the answers, but it shows much worse agreement on hard questions compared to inter-human agreements. However, harder questions are actually proposed as a contribution of this work. As the answers for such puzzles can be ``off by a hair but miss by a mile'', it might require human evaluation for the hard puzzles.
> It is also a bit confusing why the human agreement on Medium questions is much lower than on hard and easy ones.**
>
> The human-human agreement rates, sometimes as low as 87.5\% as seen in Table 2, reflect the strictness of our agreement metric. For instance, if judgements from three humans result in two 'matched' and one 'unmatched,' the agreement is considered 1/3. If the judge model agree with the 'matched' responses, its agreement with humans is 2/3. Despite this rigorous evaluation, our method still achieves a high agreement rate of 88.24\% on hard puzzles. It shows the diversity of acceptable answers and the challenging nature of our benchmark.
>
>
> **Q1. Similar datasets have existed before, as mentioned in the paper, such as RiddleSense and Oogiri. Here's an example from RiddleSense:
> My life can be measured in hours. I serve by being devoured. Thin, I am quick; Fat, I am slow. Wind is my foe. What am I? (A) paper (B) candle (C) lamp (D) clock (E) worm.
> In comparison, SPLAT has longer questions and answers. Also, the reference answers for SPLAT are open-ended, while the previous ones are all multiple-choice, which is claimed as a contribution. However, this also makes the evaluation more challenging. What makes you think open-ended is a much better option that makes it a contribution?**
>
> Not all problems are suited to multiple-choice formats, such as puzzles. While multiple-choice questions are easier to evaluate, they often simplify the problem. Besides, an open-ended format lessens the likelihood of inadvertently leading to the correct answer.
> However, the evaluation of open-ended answers is much more challenging.
> To this end, we develop a multi-turn framework that leverages LLMs as judges to address open-ended evaluation tasks. The robustness of this framework represents one of the key contributions of our work.
>
>
> **Q2. Authors claim that their methods reduce reliance on stronger evaluation models, but it seems they are still using WizardLM-2 as a component (Judge Model) of their framework?**
>
> By "reduce reliance on stronger evaluation models", we mean that the judge model in our framework does not necessarily need to be more powerful than the player model. However, it still needs to be sufficiently robust, achieving a high level of consistency with human judgements, as demonstrated in Tables 2 and 3. In traditional model-based evaluation methods (e.g., [1]), the evaluation model typically needs to be more capable than the model being evaluated since it directly grades the responses.
> This requirement often constrains the ability to evaluate newer, more advanced models. Our approach, by contrast, allows for the use of robust but not necessarily superior models as evaluators, facilitating a broader assessment of SoTA models.
>
> [1] Judging llm-as-a-judge with mt-bench and chatbot arena. NeurIPS, 2023.
>
>
> **L1. The size for this benchmark is small compared with previous similar datasets that have existed before.**
>
> Please refer to General Response **G2**.

---

### Official Review · Reviewer_6oC8 · 2024-07-03

**Soundness:** 3
**Presentation:** 3
**Contribution:** 3
**Rating:** 7
**Confidence:** 4

**Summary:**

This paper focuses on lateral thinking, which is about creativity and viewing problems from multiple angles. To sovle the challenge that the complexity of assessing creative thought processes and the scarcity of relevant data, this paper introduces SPLAT, a benchmark leveraging Situation Puzzles to evaluate and elicit LAteral Thinking of LLMs. By employing a new multi-turn player-judge framework instead of the traditional model-based evaluation, the proposed method lessens dependence on more robust evaluation models, enabling the assessment of state-of-the-art LLMs. Moreover, through applying data and reasoning processes from our benchmark to another lateral thinking-related benchmark, like RiddleSense, leads to performance enhancements.

**Strengths:**

1. This paper focuses on the lateral thinking problem, which is important to creativity and viewing problems from multiple angles.
2. The proposed dataset is open-ended and the evaluation method is good.
3. The paper is well-written.

**Weaknesses:**

1. The size of the dataset is samll.
2. The reference answer is limited to 1, while the questions are one-to-many questions.
3. The in-depth analysis is not enough

**Questions:**

1. Why not provide more than one answer for each question (I do not think it is challenging, there are multiple answers to a question on the internet such as Quora)? Since one such question could have multiple answers, more reference answer might also be helpful for evaluation.
2. A overall evaluation metric is needed, some model might have a high acc with a high average round, you should define a overall metric to combine these two metrics for a clear comparison of existing LLMs.
3. It is better to adopt more benchmarks for validating the effectiveness of the proposed dataset (more than just RiddleSense).

---

> ### Author Rebuttal · Authors · 2024-08-07
>
> **W1. The size of the dataset is small.**
>
> Please refer to General Response **G2**.
>
> **W2+Q1. The reference answer is limited to 1, while the questions are one-to-many questions. Why not provide more than one answer for each question (I do not think it is challenging, there are multiple answers to a question on the internet such as Quora)? Since one such question could have multiple answers, more reference answers might also be helpful for evaluation.**
>
> We have tried to directly gather questions and answers from question-and-answer websites like Quora, where we observed that situation puzzle queries often ask "How to build a situation puzzle?" with responses typically listing several one-to-one puzzle pairs rather than multiple answers to a single question.
>
> Thus, as discussed in the paper, we originally wanted to evaluate it like in MS COCO, where multiple reference answers are provided for each image, and metrics are calculated based on the closest reference answer. However, due to the difficulty in collecting multiple reference scenarios for situation puzzles, we adapt our approach. Instead, we run each puzzle multiple times (denoted as $R$) and select the closest match to the reference scenario from these iterations as the final result. This method preserves the diversity of potential answers and mitigates the effects of hallucinations in LLMs by allowing multiple evaluations per puzzle.
>
> In the future, we seek to employ crowd-sourcing platforms like Amazon Mechanical Turk to collect diverse answers to the same puzzle. Besides, we will provide contributors with guidelines that encourage diverse thinking, asking them to provide unique answers or perspectives on the same puzzle.
>
> **W3+Q3. The in-depth analysis is not enough. It is better to adopt more benchmarks for validating the effectiveness of the proposed dataset (more than just RiddleSense).**
>
> Please refer to General Response **G1**.
>
> **Q2. An overall evaluation metric is needed, some models might have a high acc with a high average round, you should define an overall metric to combine these two metrics for a clear comparison of existing LLMs.**
>
> Thank you for your constructive suggestion. Based on the calculation of Accuracy (Acc, \%) and Average Round (Rnd) in our paper, we further define an overall evaluation metric as Overall = $1/N\sum_{i=1}^N\mathbb{I}$(sample$_i$) / Rnd$_i$, where $\mathbb{I}$ is an indicator function that returns 1 if the scenario deduced by LLMs matches the reference scenario/answer semantically, and 0 otherwise. Given that $\mathbb{I}$(sample$_i$) takes values in \{0, 1\} and Rnd$_i$ ranges from 1 to a maximum (maximum = 15 in our paper), the Overall metric spans from 0 to 1, with higher values indicating better performance.
>
> The table below reflects trends similar to those discussed in our paper, where GPT-4 and its Turbo variant, as well as WizardLM-2 (8x22B) show strong performance and outperform other models. GPT-4 leads with an Overall score of 8.16 (Avg, x100), indicating high capability. However, it is still far from saturation, which suggests room for further improvement.
>
> | Model | Overall (Easy, x100) | Overall (Medium, x100) | Overall (Hard, x100) | Overall (Avg, x100) |
> | --- | --- | --- | --- | --- |
> | LLama3-8B | 3.65| 1.58 | 0.50 | 1.91 |
> | LLama3-70B  | 8.67  | 3.54    | 1.36  | 4.52 |
> | Qwen1.5-32B| 6.39  | 2.47    | 2.01  | 3.62 |
> | Qwen1.5-110B | 9.71  | 3.74    | 2.16  | 5.20  |
> | WizardLM-2-8x22B | 11.97 | 4.67    | 2.74  | 6.46 |
> | GPT-4 Turbo | 13.22 | 5.06    | 1.35  | 6.54 |
> | GPT-4  | 15.25 | 6.71    | 2.52  | 8.16 |

---

> > ### Comment · Reviewer_6oC8 · 2024-08-12
> > **Replying to Rebuttal of Authors**
> >
> > Thanks for your detailed response. It has addressed some of my concerns, I will raise the score to reflect this.
> > However, I am still concerned that the number of answers to each question is limited to 1 since your paper is about "Lateral Thinking". Although "How to build a situation puzzle?" is not proper to obtain multiple answers, there still are many questions that have answers from multiple angles, such as "how to prove 1+1=2".

---

> > > ### Author Response · Authors · 2024-08-13
> > > **Thanks**
> > >
> > > Thank you for your valuable suggestions and for increasing the score. We will explore incorporating multi-answer formats in future updates to better evaluate lateral thinking in LLMs.

---

### Official Review · Reviewer_ZJ8a · 2024-07-12

**Soundness:** 2
**Presentation:** 3
**Contribution:** 3
**Rating:** 5
**Confidence:** 3

**Summary:**

This work focuses on creating a benchmark, as well as a modeling framework for evaluating lateral thinking of LLMs. The benchmark, called SPLAT, consists of 975 situation puzzles. The framework consists of a “judge” and a “player” (the LLM to be evaluated). The judge poses an open ended puzzle that is further clarified by the player till a final answer is reached. The judge evaluates the final answer. The authors posit that this framework eases the challenge faced with typical auto-evaluation models since the judge does not need to be better than the LLM that is being evaluated.

The authors evaluate the quality of the judge (via human agreement), and finally evaluate LLMs using a promising judge.

**Strengths:**

The motivation for having a lateral thinking benchmark is strong. The work around automatic evaluation using a judge v/s player framework is well thought out. I think such a benchmark as well as framework/protocol would be beneficial to the community.

**Weaknesses:**

Overall, the idea and motivation behind the work is solid. I also think that the benchmark is good quality. However, my concern is that this work is half-baked. The experiments aren’t conclusive. More work should go into making the experiments thorough and useful to the scientific community.


* Although the motivation and overall idea is strong, I am concerned about the experimental setup. The agreement rates for WizardLM-2 in Table 2 and 3 don’t appear to be high enough (are around ~80%) and 3 individuals seems to be fairly less. How does this fare with human agreement rates in literature? Also, is there any reason we haven’t used the strongest model (GPT-4 from the paper) as the judge?

* I also am concerned about Table 4. The authors should consider using strong base models for evaluation (that are closer to GPT-4 on public benchmarks)

* It would be good to introduce modeling methods to improve upon the metrics that are obtained for models on SPLAT. This will give more insight into how this benchmark is valuable to the community.

* nit: I see mentions of “using data and reasoning processes from our framework” consistently throughout the paper. This should be framed better and be more precise.

**Questions:**

See weaknesses mentioned above.

---

> ### Author Rebuttal · Authors · 2024-08-07
>
> **W1. Although the motivation and overall idea are strong, I am concerned about the experimental setup. The agreement rates for WizardLM-2 in Tables 2 and 3 don’t appear to be high enough (are around ~80\%) and 3 individuals seem to be fairly less. How does this fare with human agreement rates in literature?**
>
> Thank you for pointing out this. Due to the strict calculation method of our agreement, the agreement between humans cannot always reach 100\%.
> Specifically, as mentioned in the supplementary. if three humans vote 'matched', 'matched', and 'unmatched' for a puzzle, respectively, the agreement among them, noted as 'human-human', is only 1/3, as there are three pairs: (matched, matched), (matched, unmatched), and (matched, unmatched). If the judge model votes `matched', the agreement between humans and the judge model is 2/3.
>
> But even under such a strict evaluation metric, the judge model WizardLM-2 (8x22B) aligns closely with human judgements, achieving over 80\% agreement on both final answers and reasoning processes. Specifically, as shown in Table 2, the final answer agreements between WizardLM-2 (8x22B) and human evaluations are 100\%, 87.50\%, and 100\% for easy, medium, and hard puzzles, respectively. Similarly, the agreement on reasoning processes (Table 3) shows robust results with 84.18\%, 89.86\%, and 90.15\% for each difficulty level, respectively.
>
> **W2. Also, is there any reason we haven’t used the strongest model (GPT-4 from the paper) as the judge?
> I also am concerned about Table 4. The authors should consider using strong base models for evaluation (that are closer to GPT-4 on public benchmarks).**
>
> Before the submission deadline, data from its official website [1] indicates that WizardLM-2 (8x22b) performs competitively on benchmarks like MT-Bench, comparable to leading models such as GPT-4, i.e., WizardLM-2 (8x22b): 9.12 vs. GPT-4-0314: 8.96 vs. GPT-4-1106-Preview: 9.32. Besides, we also conduct an experiment that employ GPT-4 as the judge model, which yields results comparable to WizardLM-2 in both final answer agreement and reasoning process agreement. For instance, on "Hard" puzzles, GPT-4 achieved an 89.85\% final answer agreement rate, closely matching WizardLM-2's 88.24\%.
>
> [1] https://wizardlm.github.io/WizardLM2/
>
> **W3. It would be good to introduce modeling methods to improve upon the metrics that are obtained for models on SPLAT. This will give more insight into how this benchmark is valuable to the community.**
>
> To enhance the alignment/agreement between the judge model and human assessments, we plan to focus on two approaches in future work: 1) Employing in-context learning methods such as Chain-of-Thought (CoT) [2] or Tree of Thought (ToT) [3], which follow human-like reasoning patterns and thus can improve alignment; 2) Utilising training-based methods like Supervised Fine-Tuning (SFT) and Direct Preference Optimisation (DPO) [4], which involve training models directly on lateral thinking puzzles to enhance their accuracy in evaluating such scenarios. These methods, while promising, are beyond the scope of this paper. We regard them as our future work.
>
> [2] Chain-of-Thought Prompting Elicits Reasoning in Large Language Models. NeurIPS, 2022.
>
> [3] Tree of Thoughts: Deliberate Problem Solving with Large Language Models. NeurIPS, 2023.
>
> [4] Direct Preference Optimisation: Your Language Model is Secretly a Reward Model. NeurIPS, 2023.
>
> **W4. nit: I see mentions of "using data and reasoning processes from our framework" consistently throughout the paper. This should be framed better and be more precise.**
>
> This refers to 'using the question-answer pairs generated during our SPLAT's player-judge interactions as auxiliary prompts'. We will revise this thoroughly and make it clearer.

---

> > ### Comment · Reviewer_ZJ8a · 2024-08-13
> >
> > Thanks, I think W1 and W2 make sense. I have increased the score to reflect this. I do think W3 would be important to improve the potential impact of this work.

---

> > > ### Author Response · Authors · 2024-08-13
> > > **Thanks**
> > >
> > > Thank you again for the constructive comments. We will add the clarifications in the response to our revised paper.

---

### Official Review · Reviewer_s9Dk · 2024-07-17

**Soundness:** 4
**Presentation:** 3
**Contribution:** 4
**Rating:** 8
**Confidence:** 5

**Summary:**

This paper introduces SPLAT, a novel benchmark for evaluating and eliciting lateral thinking abilities in Large Language Models (LLMs) using situation puzzles. The key contributions include: A new dataset of 975 graded situation puzzles across three difficulty levels.
A multi-turn player-judge evaluation framework that reduces reliance on stronger evaluation models. Demonstration that the benchmark can both evaluate and elicit lateral thinking in LLMs. Experimental results showing that a robust evaluation model (WizardLM-2) closely matches human judgments. Evidence that applying SPLAT's data and reasoning processes to other lateral thinking benchmarks leads to performance improvements in LLMs.

**Strengths:**

1. The paper presents a novel approach to assessing lateral thinking in LLMs, an area that has been underexplored compared to vertical thinking. The use of situation puzzles and the multi-turn player-judge framework are creative solutions to the challenges of evaluating open-ended, creative thought processes.
2. The methodology appears rigorous, with careful data collection, annotation, and difficulty categorization. The multi-turn player-judge framework is well-designed to overcome limitations of traditional model-based evaluations. The experimental results, including comparisons with human judgments and applications to other benchmarks, provide strong evidence for the effectiveness of the approach.
3. The paper is well-structured and clearly written. The task definition, data construction process, and evaluation framework are explained in detail. Figures and tables effectively illustrate key concepts and results.
4. This work addresses an important gap in the evaluation of LLMs by focusing on lateral thinking, which is crucial for creative problem-solving and innovation. The benchmark has potential applications beyond evaluation, as demonstrated by its ability to improve LLM performance on other lateral thinking tasks. This could lead to advancements in developing more creative and flexible AI systems.

**Weaknesses:**

1. While the data collection process is described, there's minimal discussion of potential biases in the dataset, such as cultural specificity of the puzzles or biases introduced during the human annotation process.
2. The paper would benefit from ablation studies to isolate the impact of different components of the SPLAT framework, such as the difficulty categorization or the multi-turn questioning process.
3. The paper lacks a discussion of potential ethical implications or misuse of improved lateral thinking capabilities in LLMs.

**Questions:**

1. How does the performance of LLMs on SPLAT correlate with their performance on other, more traditional NLP tasks? Is there evidence that lateral thinking ability is distinct from other language model capabilities?
2. Have you considered ways to automatically generate new situation puzzles to expand the dataset and improve scalability?
3. How sensitive is the performance of LLMs on SPLAT to the specific prompts or instructions given? Could you elaborate on the prompting strategy used?

**Limitations:**

None.

---

> ### Author Rebuttal · Authors · 2024-08-07
>
> **W1. There's minimal discussion of potential biases in the dataset.**
>
> During our data collection process, we also track user preferences for each situation puzzle, measured by a preference score (\%). We notice a pattern where puzzles (about 1\% of our dataset) with lower preference rates (30\%-40\%) show greater variation in the time taken to solve them, ranging from 5 to 26 minutes for puzzles categorised as "Medium". In contrast, puzzles with higher preference rates (above 80\%) exhibit less time variation, with solving times ranging from 9 to 17 minutes for "Medium" puzzles. This suggests that annotations may introduce more noise when users engage with puzzles they find less appealing.
>
> To avoid this bias, we will enhance the diversity of the puzzles to ensure a broader appeal across different user groups, thereby reducing the likelihood of low preference scores. Besides, we plan to implement a dynamic refinement method during data collection to enhance puzzle engagement. If a puzzle receives a low preference score, we will revise it based on user feedback and re-release it to assess improvements in user preference.
>
> **W2. Ablation studies to isolate the impact of different components of the SPLAT framework.**
>
> We conduct an ablation study to explore the influence of our data and reasoning processes. In addition to the RiddleSense, we include the BrainTeaser benchmark, which features both word and sentence puzzles. As shown in the table below, incorporating our data into base LLMs leads to an increase in average accuracy (Llama3-8B: 61.45 -> 64.07; Llama3-70B: 80.76 -> 83.78). This improvement is further enhanced to 66.71 and 85.42, respectively, when we integrate the reasoning processes. These results further demonstrate that both our data and reasoning processes can effectively enhance the lateral thinking capabilities of LLMs.
>
> | Model | RiddleSense (Dev) | BrainTeaser (Sentence, overall) | BrainTeaser (Word, overall) | Average  |
> | ------- | ------- | ------- | ------- | ------- |
> | Llama3-8B (base) | 70.51 | 67.65    | 46.2 | 61.45 |
> | Llama3-8B (data)| 70.32 | 69.62    | 52.28| 64.07 |
> | Llama3-8B (data+reasoning)  | 72.18 | 69.89    | 58.07| 66.71 |
> | Llama3-70B (base)| 83.34 | 87.76    | 71.2 | 80.76 |
> | Llama3-70B (data) | 82.95 | 91.12    | 77.27| 83.78 |
> | Llama3-70B (data+reasoning) | 85.21 | 91.51    | 79.54| 85.42 |
>
> **W3. Potential ethical implications or misuse.**
>
> Thank you for the kind reminder. We have discussed these in Section A.1 of our supplementary. Briefly, while enhancing LLMs in lateral thinking offers numerous benefits, it also introduces potential risks such as the misuse of technology for fraud or misinformation. Thus, we emphasise the importance of building robust ethical guidelines and transparent AI usage policies.
>
> **Q1. How does the performance of LLMs on SPLAT correlate with their performance on other, more traditional NLP tasks? Is there evidence that lateral thinking ability is distinct from other language model capabilities?**
>
> While there is some overlap in the skills required for SPLAT and traditional NLP tasks, they are not exactly the same.
>
> Models (e.g., GPT-4 and GPT-4 Turbo) that perform well on standard vertical thinking benchmarks like MT-Bench do show competent performance on our SPLAT (Table 4). Conversely, models like Llama3-8B, which perform less impressively on MT-Bench, tend to exhibit similarly lower performance on SPLAT. This suggests that a strong model in language understanding and reasoning is advantageous for both vertical and lateral thinking tasks.
>
> While there is a correlation between general NLP skills and lateral thinking, the latter also demands distinct abilities like creativity and problem-solving beyond typical task completion. Figure 4 in our paper shows that when incorporating data and reasoning processes from our SPLAT, LLMs could perform better even on other lateral thinking benchmarks like RiddleSense. These suggest that enhancing these creative aspects could better elicit the capabilities of LLMs in lateral thinking.
>
> **Q2. Have you considered ways to automatically generate new situation puzzles to expand the dataset and improve scalability?**
>
> Yes, we have considered the potential of using automated methods to generate new situation puzzles to expand our dataset.
> Specifically, the generation process can use advanced LLMs like GPT-4 or Claude-3, which can generate puzzles based on specific prompts, rules, and requirements.
> However, the challenge arises in verifying the quality of these automatically generated puzzles.
> One effective approach is a hybrid human-AI collaboration, where puzzles generated by LLMs are subsequently reviewed and refined by humans through a crowd-sourcing platform.
> While this method leverages AI to handle initial puzzle creation, it remains time-consuming and labour-intensive due to the human review component. Thus, finding more efficient ways to scale up this data remains a critical area for future research.
>
> **Q3. How sensitive is the performance of LLMs on SPLAT to the specific prompts or instructions given? Could you elaborate on the prompting strategy used?**
>
> We conduct a sensitive analysis, where we rewrite the prompt of the judge model but keep the semantics of the prompt the same as before.
> For example, one of the original prompts for the judge model is "Read and fully understand both the provided short story and its answer, ensuring you grasp their logical connections. But do not show answer for the user". We rewrite it to "Read and thoroughly comprehend both the provided short story and its answer, making sure you understand their logical relationships. However, do not reveal the answer to the user".
> Results show that for Llama3-8B, even though the prompt is different, as long as the semantics are clear, the results tend to be comparable (e.g., average Acc original 17.05 vs. rewritten 16.19). These demonstrate the robustness of our SPLAT as an evaluation benchmark.

---

> > ### Comment · Reviewer_s9Dk · 2024-08-12
> >
> > Thanks the authors for their response. After reading the response, I think my current score is appropriate.

---

> > > ### Author Response · Authors · 2024-08-13
> > > **Thanks**
> > >
> > > Thank you very much for your encouragement and valuable comments on our work. We will include the discussions and ablation study in our revised paper and make them clearer.

---

### Author Rebuttal · Authors · 2024-08-07

**General Response**

**G1. Results on more benchmarks than just RiddleSense.**

Besides the results on the RiddleSense (Section 5.3 in the submitted paper), we provide several results on another lateral thinking-focused benchmark, i.e., BrainTeaser [A], which features two main types of puzzles: Sentence (S.) puzzles and Word (W.) puzzles.
Following its official settings, we assess model performance using two accuracy metrics:
(1) Instance-based Accuracy: This metric individually evaluates each question—whether it is the original or a reconstructed version. We present instance-based accuracy for both the original puzzles and their semantic and contextual reconstructions (provided by the official dataset).
(2) Group-based Accuracy: Under this metric, each original puzzle and its variants are treated as a group. The model earns a score of 1 only if it correctly solves all three puzzles within a group. If it fails to do so, the score awarded is 0.

As in our submitted paper, we consider both open-source and closed-source LLMs, specifically Llama3 (in its Instruct versions) and GPT-4, with additional considerations for different model sizes, such as Llama3-8B and Llama3-70B.
As shown in the following tables, in the zero-shot setting, the accuracy of various LLMs consistently improves on the BrainTeaser benchmark (both sentence and word puzzles). The results further demonstrate that the data and framework of our benchmark effectively elicit lateral thinking capabilities in LLMs when applied to various lateral thinking tasks.

[A] Brainteaser: Lateral thinking puzzles for large language model. EMNLP, 2023.

BrainTeaser (Sentence)

| Model| Original | Semantic| Context | Ori \& Sem | Ori \& Sem \& Con | Overall |
| --- | --- | --- | --- | --- | --- | --- |
| Llama3-8B| 70.23 | 63.09 | 69.64 | 52.09 | 38.32 | 67.65 |
| Llama3-8B*| 72.18 | 65.47 | 72.02 | 58.92 | 45.23 | 69.89 |
| Llama3-70B | 89.34 | 87.57 | 86.39 | 84.61 | 75.73 | 87.76 |
| Llama3-70B*| 93.49 | 90.53 | 90.53 | 88.75 | 82.84 | 91.51 |
| GPT-4| 93.49 | 89.94 | 83.43 | 88.75 | 75.14 | 88.95 |
| GPT-4*| 95.26 | 91.71 | 88.69 | 91.71 | 82.24 | 91.88 |

BrainTeaser (Word)

| Model| Original | Semantic| Context | Ori \& Sem | Ori \& Sem \& Con | Overall |
| --- | --- | --- | --- | --- | --- | --- |
| Llama3-8B| 47.72 | 44.69 | 46.21 | 31.06 | 17.42 | 46.20 |
| Llama3-8B*| 54.54 | 54.54 | 65.15 | 39.39 | 29.54 | 58.07 |
| Llama3-70B| 71.96 | 71.96 | 69.69 | 62.87 | 49.24 | 71.20 |
| Llama3-70B*| 81.06 | 81.81 | 75.75 | 74.24 | 59.09 | 79.54 |
| GPT-4| 71.21 | 65.91 | 56.06 | 59.09 | 41.66 | 64.39 |
| GPT-4*| 74.24 | 72.72 | 62.12 | 64.39 | 45.45 | 69.69 |


**G2. The size of the dataset.**

Our dataset contains 975 puzzles, where the number of puzzles is more than that in BrainTeaser (Sentence) and BrainTeaser (Word), which are 627 and 492, respectively. Besides, as each of our puzzles requires multi-turn conversations (always more than 5) to solve, it is comparable to the dataset like RiddleSense or Oogiri (T2T, Eng.) in terms of inference volume, where both contain about 5,000 puzzles. Moreover, as a benchmark, our priority is to ensure diversity and robustness to produce a stable evaluation framework. We believe the current quantity of puzzles is sufficient for this purpose. We also acknowledge the need for a larger dataset and plan to expand SPLAT in future versions.

---

### Decision · Program_Chairs · 2024-09-25

**Decision:**

Accept (poster)

**Comment:**

This paper introduces a novel benchmark for measuring the strength of LLMs. In particular the new benchmark captures the "lateral thinking" which I think is similar to some form of common sense reasoning that is not very vertical specific like current benchmarks on QA, Math abilities etc. The evaluation proposes a new multi-turn player-judge framework instead of the traditional model-based evaluation. The proposed method lessens dependence on more robust evaluation models, enabling the assessment of state-of-the-art LLMs. The paper is well-written and presents a novel way of doing evaluation of LLMs. The authors also show that improving the model in a few shot setting to perform better on their benchmark also improves the performance of the model on another related benchmarks like RiddleSense displaying the generalizability of the model properties while hilclimbing on SPLAT.